# Inhibition of NMDA Receptor Activation in the Rostral Ventrolateral Medulla by Amyloid-β Peptide in Rats

**DOI:** 10.3390/biom13121736

**Published:** 2023-12-02

**Authors:** Md Sharyful Islam, Chih-Chia Lai, Lan-Hui Wang, Hsun-Hsun Lin

**Affiliations:** 1Master and Ph.D. Programs in Pharmacology and Toxicology, School of Medicine, Tzu Chi University, Hualien 97004, Taiwan; 107752102@gms.tcu.edu.tw; 2Department of Pharmacology, School of Medicine, Tzu Chi University, Hualien 97004, Taiwan; cclai@mail.tcu.edu.tw; 3Department of Pharmacy, Buddhist Tzu Chi General Hospital, Buddhist Tzu Chi Medical Foundation, Hualien 97004, Taiwan; 4Department of Physiology, School of Medicine, Tzu Chi University, Hualien 97004, Taiwan; lanhuipl@mail.tcu.edu.tw

**Keywords:** amyloid-beta, cardiovascular function, RVLM, NMDA receptors, GluN2B, phosphorylation

## Abstract

N-methyl-D-aspartate (NMDA) receptors, a subtype of ionotropic glutamate receptors, are important in regulating sympathetic tone and cardiovascular function in the rostral ventrolateral medulla (RVLM). Amyloid-beta peptide (Aβ) is linked to the pathogenesis of Alzheimer’s disease (AD). Cerebro- and cardiovascular diseases might be the risk factors for developing AD. The present study examines the acute effects of soluble Aβ on the function of NMDA receptors in rats RVLM. We used the magnitude of increases in the blood pressure (pressor responses) induced by microinjection of NMDA into the RVLM as an index of NMDA receptor function in the RVLM. Soluble Aβ was applied by intracerebroventricular (ICV) injection. Aβ1-40 at a lower dose (0.2 nmol) caused a slight reduction, and a higher dose (2 nmol) showed a significant decrease in NMDA-induced pressor responses 10 min after administration. ICV injection of Aβ1-42 (2 nmol) did not affect NMDA-induced pressor responses in the RVLM. Co-administration of Aβ1-40 with ifenprodil or memantine blocked the inhibitory effects of Aβ1-40. Immunohistochemistry analysis showed a significant increase in the immunoreactivity of phosphoserine 1480 of GluN2B subunits (pGluN2B-serine1480) in the neuron of the RVLM without significant changes in phosphoserine 896 of GluN1 subunits (pGluN1-serine896), GluN1 and GluN2B, 10 min following Aβ1-40 administration compared with saline. Interestingly, we found a much higher level of Aβ1-40 compared to that of Aβ1-42 in the cerebrospinal fluid (CSF) measured using enzyme-linked immunosorbent assay 10 min following ICV administration of the same dose (2 nmol) of the peptides. In conclusion, the results suggest that ICV Aβ1-40, but not Aβ1-42, produced an inhibitory effect on NMDA receptor function in the RVLM, which might result from changes in pGluN2B-serine1480 (regulated by casein kinase II). The different elimination of the peptides in the CSF might contribute to the differential effects of Aβ1-40 and Aβ1-42 on NMDA receptor function.

## 1. Introduction

Central N-methyl-D-aspartate (NMDA) receptors play a crucial role in regulating physiological processes, including excitatory synaptic transmission and synaptic plasticity [1]. However, NMDA receptor dysfunction has been linked to the pathological mechanism of neurodegeneration [1]. Alzheimer’s disease (AD) is a complex neurodegenerative disorder and the leading cause of dementia [2]. Most scientists agree that two proteins are heavily involved in AD: amyloid-beta (Aβ) and tau protein. The accumulation of Aβ in the brain of AD patients leads to plaques that clump between neurons and disrupt normal cell functioning. The tau protein causes neurofibrillary tangles inside neurons, thereby blocking the neuronal transport system [3]. NMDA receptor response is the most decisive factor linking Aβ accumulation and glutamate toxicity. Soluble Aβ oligomers bind to GluN1 and GluN2B subunits, thereby reducing the surface expression of NMDA receptors, causing NMDA receptor dysfunction and further leading to dendritic spine destruction, which is the main cause of AD-induced nerve fiber network damage [4]. 3-((D)-2-carboxypiperazin-4-yl)-propyl-1-phosphonic acid, a competitive NMDA receptor antagonist, prevented the effect of Aβ oligomers on spine density [5]. Further, evidence shows that soluble oligomeric forms of Aβ may interrupt synaptic function and are associated with early neurodegeneration in AD, primarily due to modifications to the glutamatergic system [6,7]. On the other hand, the accumulation of Aβ plaques may result in the oxidative stress elements involved in AD, which plays a central role in the pathogenesis of AD [8,9]. The impact of Aβ accumulation on postsynaptic glutamate AMPA/NMDA receptors is well documented. Applying Aβ has decreased the amplitude and frequency of AMPA postsynaptic currents in CA1 pyramidal neurons [10]. Aβ regulates the NMDA receptors and disturbs the ionic balance between synaptic and extrasynaptic NMDA receptor signaling [11]. Synthetic Aβ1-42 inhibited not only NMDA receptor-dependent long-term potentiation (LTP) but also voltage-activated Ca^2+^-dependent LTP induced using intense conditioning stimulation during NMDA receptor blockade [12]. This finding is consistent with evidence showing that Aβ oligomers such as Aβ1-42 can affect the surface expression of NMDA receptors and decrease NMDA receptors conductance in cortical and hippocampal neurons [13,14]. Alternatively, in vivo studies indicate that the level of NMDA receptor activation may control Aβ production; low levels of NMDA receptor activation increase, and higher levels decrease Aβ production [15]. In vitro studies also showed increased NMDA receptor conductance by Aβ oligomers in the hippocampus via mechanisms involving GluN2B subunit phosphorylation and protein kinase C (PKC) signaling pathway [16]. A study using an oocyte expression system found that Aβ1-42 oligomers may directly activate NMDA receptors, particularly with the GluN2A subunit [17]. Thus, the effects of Aβ oligomers on NMDA receptor function might differ depending on different fragments of oligomers, tissue, and experimental conditions. Memantine, an NMDA receptor channel blocker, is currently used as an anti-AD drug for the treatment of mild to moderate AD by acting as an open-channel blocker [18,19]. The activation of extrasynaptic NMDA receptors can lead to cell death and is also considered a contributing factor in the development of AD, which is selectively blocked by memantine [20].

Cerebrovascular and cardiovascular diseases may be one of the major risk factors for developing AD [21,22]. Evidence demonstrates that vascular dysfunction linked between major risk factors hypertension and apolipoprotein E_4_ may lead to cerebrovascular and cardiovascular disorders associated with AD [23,24]. A recent cohort study showed the association of higher late-life blood pressure with an increasing number of brain infarcts and AD neuropathology [25]. Solubilized Aβ has been found in both in vitro and in vivo vasoactive effects on cerebral vessels. Additionally, Aβ can potentially enhance the activity of both exogenous and endogenous vasoconstrictors and may directly affect the contraction of cerebral vessels, which may contribute to cerebrovascular dysfunction [26]. Thus, the mechanisms of AD-related vascular disease might involve a combination of vascular and neuronal toxicity [27].

The rostral ventrolateral medulla (RVLM) is well-known as the pressor region of the medulla in the brainstem area, which is responsible for basal and reflex control of sympathetic activity associated with cardiovascular function [28]. Several cardiovascular diseases, such as heart failure and hypertension, are associated with the abnormally elevated level of sympathetic activity in the RVLM [28]. NMDA receptors are essential in regulating sympathetic nerve activity and cardiovascular function. Glutamate is the primary neurotransmitter in the RVLM. Studies show that activating glutamate NMDA receptors in the RVLM will increase sympathetic tone and blood pressure. [29,30]. Phosphorylation of NMDA receptor subunits can change the activity of NMDA receptors. Many kinases can phosphorylate NMDA receptors, including protein kinase A, PKC, protein tyrosine kinase, casein kinase II (CKII), etc [31]. It is still unclear whether Aβ affects the NMDA receptor of the RVLM, mainly whether Aβ participates in cardiovascular regulation by changing the NMDA receptor phosphorylation state in the RVLM. We perform in vivo experiments by intracerebroventricular (ICV) injection of Aβ1-40 or Aβ1-42 or combined administration of Aβ and NMDA receptor antagonist to evaluate the effects of these drugs on the pressor effects of microinjection of NMDA into the RVLM. The increase in blood pressure (pressor response) caused by the microinjection of NMDA into the RVLM region was used as an index of NMDA receptor function. In addition, we also used immunohistochemical staining to detect changes in the expression of NMDA receptor subunits and their phosphorylation in the RVLM region after Aβ administration. Since animals can metabolize Aβ, we measure the Aβ content in cerebrospinal fluid (CSF) after ICV administration to determine the final concentration of the peptides in CSF. Our results suggest that increases in Aβ1-40 levels in the brain might have an inhibitory effect on NMDA receptor function in the RVLM via regulating the CKII pathway and the subsequent changes in the phosphorylated status of the NMDA receptor. The different effects of different Aβ on NMDA receptor functions might be, at least partly, due to their distinct elimination in the CSF.

## 2. Materials and Methods

### 2.1. Animals

All animal care and experimental procedures were carried out according to the guidelines of the Institutional Animal Care and Use Committee of Tzu Chi University (protocol no. 108075). The animals were housed and maintained with ad libitum access to food and water under a constant 12 h light/dark cycle in the controlled room temperature at 23 ± 1 °C and 50 ± 10% humidity. Male Sprague Dawley (SD) rats weighing 300–400 g were purchased and used in this experiment (BioLASCO Co., Ltd., Taipei, Taiwan).

### 2.2. Chemicals

We purchased Aβ1-40 and Aβ1-42 (human) from Taiclone Biotech Corp. (Taipei, Taiwan). Urethane, ifenprodil, memantine, NMDA, and other chemicals were purchased from Sigma Co. (St. Louis, MO, USA). The reagent required for immunochemical staining was obtained from Vector Laboratories, Inc. (Burlingame, CA, USA). We use a Human Aβ40 and Aβ42 enzyme-linked immunosorbent assay (ELISA) kit from Invitrogen (Viena, Austria) to measure the level of Aβ1-40 and Aβ1-42, respectively, in the CSF. The ELISA kit would mainly detect soluble Aβ [32]. We obtained anti-GluN1 and anti-GluN2B antibodies from Invitrogen (Waltham, MA, USA), anti-phospho-GluN1-ser896 and anti-phospho-GluN2B-ser1480 antibodies from Bioss Inc. (Woburn, MA, USA). Biotinylated goat–anti-rabbit IgG 488 was obtained from Abcam.

### 2.3. Human Amyloid-Beta (Aβ) ELISA Assay

CSF samples (100–200 µL) were typically collected from the rats by cisterna magna puncture. We added protease inhibitors, centrifuged the collected CSF samples to remove cellular debris, and kept them in the refrigerators (−20 °C). A sandwich ELISA kit was used to quantify the level of human Aβ in rat’s CSF according to the manufacturer protocol. The prepared ELISA plates were coated with antibodies specific to human amyloid beta. In brief, we added the CSF samples diluted appropriately and standards to the wells and incubated them to allow Aβ in the samples to bind the immobilized antibodies. Then, we washed the plates to remove unbound components. We added a detection antibody specific to Aβ and then a secondary antibody conjugated to an enzyme. Finally, we added a substrate that reacts with the enzymes to produce measurable signals, which an ELISA reader detected. To validate the detection of Aβ levels in the CSF, a different dilution of the sample from CSF was performed to ensure that the sample concentration was within the range of the standard curve. The Aβ1-40 sample from CSF was diluted 1000 and 3000 times; Aβ1-42 was diluted 30 and 100 times to measure the concentration. Each sample was analyzed in duplicate. The concentration of Aβ was calculated from the standard curve.

### 2.4. Drugs Administration and Blood Pressure Measurement

The procedure for ICV administration of anesthetized rats was similar to those described in our previous study [33,34]. Adult male Sprague Dawley 8–10 weeks old rats (300–400 g) were used in this experiment under urethane anesthetized (1.2–1.5 g/kg i.p.). The femoral artery was cannulated with polyethylene tubing (PE 50) and connected with its output to BIOPAC Systems Inc. (MP100 or MP160) to measure pulsatile and mean arterial pressure. After a set of blood pressure recordings, each rat was arranged in a stereotaxic frame for ICV and RVLM microinjection using a stainless still needle connected to the Hamilton microsyringes (10 µL). Drugs were microinjected using a syringe pump (KDS 100) at a rate of 10 µL/min. The magnitude of increases in the blood pressure (pressor responses) induced by microinjection of NMDA (0.14 nmol, 100 nL) into the RVLM every 30 min was used as an index of NMDA receptor function in the RVLM. Aβ and 0.9% saline (5 µL) were applied intracerebroventricularly. The data were processed and calculated before the treatment and at a different period after drug administration.

### 2.5. Immunohistochemistry and Immunofluorescence

The procedure of immunostaining of RVLM tissue sections was similar to those previously described [34,35]. The animals were anesthetized with i.p. urethane (1.2–1.5 g/kg). The rat was sacrificed at 10 min following ICV administration of Aβ or saline by transcardial perfusion with 50 mL of saline followed by 300–400 mL of paraformaldehyde (4%) in 0.1 M phosphate buffer (pH 7.4). The RVLM was post-fixed in the same fixative overnight and then cryoprotected in 30% sucrose in 0.05 M phosphate buffer (pH 7.4) for 48 h. Coronal brainstem sections (30 μm) were cut through the segments using a cryostat (Leica CM3050S, Leica Microsystems Nussloch GmbH, Nussloch, Germany). The RVLM tissue sections were treated with 3% hydrogen peroxide to eliminate endogenous peroxidase activity and incubated in 2% normal goat serum and 0.3% Triton X-100 to block non-specific binding. Before performing immunohistochemical staining, several primary antibody dilutions were tested to verify antibody specificity. The sections were incubated in rabbit anti-GluN1 polyclonal antibody (1:500) and rabbit anti-phospho-GluN1-ser896 antisera (1:500) for 48 h at 4 °C. All sections were then incubated in biotinylated goat–anti-rabbit IgG (1:200), followed by incubation for an additional 60 min with an avidin-biotin complex solution. Following the wash, sections were reacted with a DAB substrate kit in the presence of hydrogen peroxide to enable visualization of the precipitate. The sections were washed and mounted on gelatin-subbed slides, and the slides were dried, dehydrated in alcohol (50–100% gradually), cleared in xylene, and covered with coverslips. The sections were examined under a brightfield microscope at a magnification of 100× to localize neurons in the area of the RVLM. For the relative quantification of immunoreactivity, each image field was captured using a CCD camera mounted on a microscope (Nikon Eclipse E800, Corporation, Tokyo, Japan). The immunofluorescence procedure of RVLM tissue sections was similar to those previously described. The sections were incubated in rabbit anti-GluN2B polyclonal antibody (1:200) and rabbit anti-phospho-GluN2B-ser1480 antisera (1:100) for 48 h at 4 °C. All sections were then incubated in biotinylated goat–anti-rabbit IgG 488 (1:200). The sections were mounted on antifade mounting media and covered with coverslips. The sections were examined under a confocal microscope (Confocal microscopy system (upright)_NIKON C2si+) at 100× to localize neurons in the area of the RVLM. A standardized region of interest (500 µm square side) was aligned and consecutively centered on the RVLM area. The average number of immunoreactive (IR) neurons on both sides per section was counted and averaged. We randomly took 4–6 sections per animal and calculated the average number of IR neurons representing the animal’s immunoreactivity to the antibody.

### 2.6. Statistical Data Analysis 

All values are expressed as mean ± SEM and statistically analyzed using GraphPad Prism version 9.0 for Windows (GraphPad Software, San Diego, CA, USA). We analyzed the original data for NMDA-induced increases in MAP (mmHg) before and after applications of Aβ statistically. The time-effect relationship of Aβ on NMDA-induced pressor responses was analyzed using the repeated measure one-way ANOVA followed by Bonferroni’s multiple comparison post-test. *p* ≤ 0.05 is considered statistically significant. We compared the effects of Aβ on NMDA-increased mean arterial pressure to the control and expressed them as percentage changes; the peak amplitude of NMDA-induced pressor responses before the application of Aβ is taken as control (i.e., 100%). Immunohistochemical data and ELISA data were analyzed using an unpaired *t*-test. *p* ≤ 0.05 is considered statistically significant.

## 3. Results

### 3.1. Aβ Decreased NMDA-Induced Pressor Responses in the RVLM

To determine the effects of Aβ on cardiovascular functions, we performed the intracerebroventricular injection of Aβ on NMDA receptor function in the RVLM. The resting MAP 81.37 ± 0.81 mmHg (*n* = 72) in urethane-anesthetized male SD rats was similar to our previous study [36,37]. Consecutive microinjection of NMDA (0.14 nmol, 100 nL) into the RVLM at intervals of 30 min induced an increase in MAP; the pressor responses were onset within 2 min and lasted for 10–15 min. ICV administration of Aβ1-40 at 0.2 nmol (*n* = 12) showed a slight reduction, and 2 nmol (*n* = 14) significantly inhibited NMDA-induced pressor responses in the RVLM. In contrast, ICV microinjection of Aβ1-42 at 2 nmol (*n* = 9) did not produce any noticeable changes in NMDA-induced pressor responses in the RVLM. The inhibitory effects of Aβ1-40 (2 nmol) on NMDA-induced responses occurred at 10 min and recovered at 40 min during the observation period after ICV administration. (Figure 1).

### 3.2. Ifenprodil, a GluN2B Receptor Antagonist, Attenuated the Inhibitory Effects of Aβ1-40 on NMDA-Induced Pressor Responses

To clarify the mechanism underlying the inhibitory effects of amyloid beta on NMDA-induced pressor responses into the RVLM, we examined the effects of co-administration of ifenprodil, a GluN2B receptor antagonist, with Aβ1-40. ICV injection of a lower dose of ifenprodil (0.5 nmol, n = 5) did not produce any noticeable changes in MAP into the RVLM, a higher dose of ifenprodil (2.5 nmol, n = 5) significantly inhibited NMDA-induced pressor responses in the RVLM during the observation period after administration (Figure 2A,B,E). The co-administration of ifenprodil (0.5 nmol) with Aβ1-40 (2 nmol) attenuated the inhibitory effects of Aβ1-40 (2 nmol) on NMDA-induced pressor responses in the RVLM (Figure 2C,F). On the other hand, Aβ1-42 (2 nmol) still had little effect on NMDA-induced pressor responses in the RVLM following co-administration of ifenprodil (0.5 nmol) with Aβ1-42 (2 nmol) (Figure 2D,G).

### 3.3. Memantine, an NMDA Channel Blocker, Blocked the Inhibitory Effects of Aβ1-40 on NMDA-Induced Pressor Responses

Memantine (9 nmol, *n* = 6) did not produce any noticeable changes in NMDA-induced pressor responses in the RVLM at 10 and 40 min after ICV administration (Figure 3A,D). However, the inhibitory effects of Aβ1-40 (2 nmol) on NMDA-induced pressor responses disappeared following co-administration of memantine with Aβ1-40 (*n* = 4) (Figure 3B,E). Similar to the effects of Aβ1-42 (2 nmol) alone, co-administration of memantine with Aβ1-42 did not show noticeable changes in NMDA-induced pressor responses in the RVLM (*n* = 6) (Figure 3C,F).

### 3.4. Aβ1-40 Does Not Change the Expression of PKC-Mediated GluN1 Phosphorylation at Serine 896 on the RVLM

Phosphorylation has emerged as a mechanism regulating NMDA receptor functions. In the in vitro study, we examined the effects of Aβ on the expression of NMDA GluN1 subunits (GluN1) and phosphoserine 896 on the GluN1 subunit (pGluN1-serine896) in the RVLM neurons determined by immunohistochemical staining. Ten minutes after ICV administration of saline or Aβ1-40 (2 nmol), a comparison of the immunoreactive (IR) neurons of NMDA receptor subunit GluN1 in RVLM showed no significant difference between the two groups. The number of GluN1-IR neurons was 63.2 ± 2.4 (*n* = 6) in the saline-treated group and 63.2 ± 3.8 (*n* = 6) in the Aβ-treated group (Figure 4A). Similarly, there was no significant difference in the number of neurons in pGluN1-serine896-IR between saline and Aβ groups. The number of neurons in the former is 48.9 ± 1.8 (*n* = 6), and the number in the latter is 46.4 ± 0.8 (*n* = 6) (Figure 4B).

### 3.5. Aβ1-40 Increased the Expression of CKII-Mediated GluN2B Phosphorylation at Serine 1480 on the RVLM

We examined the effects of Aβ on the expression of NMDA GluN2B subunits and phosphoserine 1480 on the GluN2B subunit (pGluN2B-serine1480) in the RVLM neurons determined using immunofluorescence (IF). ICV injection of Aβ1-40 (2 nmol) had little effect on the number of immunofluorescence (IF) neurons of GluN2B in the RVLM at 10 min following administration compared to saline. Aβ1-40 significantly increases the number of neurons of pGluN2B-serine1480 in the RVLM following administration (Figure 5A,B). The number of GluN2B-IF neurons was 55.2 ± 1.4 (*n* = 7) in the saline-treated group and 59.2 ± 2.7 (*n* = 7) in the Aβ-treated group (Figure 5A). There was a significant difference in the number of neurons in pGluN2B-serine1480-IF between saline and Aβ groups. The number of neurons in the former is 45.9 ± 1.0 (*n* = 8), and the number in the latter is 52.6 ± 2.0 (*n* = 6) (Figure 5B).

### 3.6. The Concentration of Aβ1-40 in CSF Is Significantly Higher Than Aβ1-42 after ICV Injection of the Same Dose of Two Peptides

To clarify the working concentration of Aβ1-40 and Aβ1-42 on NMDA receptor functions following ICV administration, we determined the human Aβ levels in rat CSF, which was quantified using ELISA techniques. The data demonstrated significantly higher levels of Aβ1-40 compared to Aβ1-42 in 10 min following ICV administration of the same dose (2 nmol) of the peptides. After treatment with Aβ1-40 and Aβ1-42, the individual peptide concentrations in the CSF were 6.3 ± 1.3 nM (*n* = 5) and 67.7 ± 17.2 nM (*n* = 5), respectively (Figure 6).

## 4. Discussion

In our previous study, we observed that different fragments of Aβ may have differential effects on the NMDA receptor function, and the specific augmentation of NMDA receptor function by Aβ1-40 may involve PKC-dependent mechanisms in sympathetic preganglionic neurons [38]. Additionally, we found that the modulation of NMDA receptors function in the RVLM might participate in a consequent pressure response [39]. Activation of NMDA receptors is required for normal brain functions, but abnormal NMDA receptor activation is associated with pathophysiology in numerous neurodegenerative disorders [40,41]. In the RVLM, glutamate is the main excitatory neurotransmitter that activates NMDA receptors, which are involved in the regulation of sympathetic tone, blood pressure, and cardiovascular functions [42,43]. In this experiment, we aim to examine the acute effects of Aβ on the function of NMDA receptors in the RVLM, a crucial region involved in regulating cardiovascular functions in CNS. Our results showed ICV Aβ1-40, but not Aβ1-42 inhibits the pressor effects induced by microinjection of NMDA into the RVLM. Inhibition of NMDA-induced pressor response by Aβ1-40 may modulate the activity of NMDA receptors in the RVLM, possibly through binding to specific receptor sites. Our study showed that Aβ1-42 had no significant effects on NMDA-induced pressor responses, suggesting Aβ1-42, at the dose used, might not directly modulate NMDA receptor-mediated blood pressure response in the RVLM. The lack of the impact of Aβ1-42 on NMDA receptor function might result from the pharmacokinetic mechanisms, as demonstrated by the results showing much lower CSF levels of the peptide compared to that of Aβ1-40.

Then, we explore the mechanism underlying the inhibitory effects of Aβ1-40 on NMDA receptor function. We co-administered Aβ with ifenprodil (GluN2B receptor antagonist) or memantine (ion channel blocker). Ifenprodil was found to be a potent negative allosteric modulator that selectively inhibits NMDA receptors containing the GluN2B subunit [44,45]. We found that co-application of Aβ1-40 with ifenprodil attenuated the inhibitory effects of Aβ1-40 on NMDA-induced pressor responses. These results showed that the inhibitory effects of Aβ1-40 might be attenuated by blocking the activation of NMDA receptors containing the GluN2B subunit, suggesting that the GluN2B subunit might play a mechanistic role in Aβ1-40 effects on NMDA receptor responses in the RVLM. Ifenprodil itself may affect the cardiovascular system, contributing to its attenuating effect on the inhibitory effects of Aβ1-40. Our results showed that ICV administration of a higher dose of ifenprodil inhibited NMDA-induced pressor responses in the RVLM. Thus, the present study used a certain amount of ifenprodil, causing no changes in NMDA receptor function. In contrast to Aβ1-40, co-administration of the same dose of Aβ1-42 with ifenprodil did not significantly affect NMDA-induced pressor responses in the RVLM. Aβ1-42 might have multiple (potentiated and inhibited) effects on NMDA receptor functions through differential mechanisms, causing no changes observed. Whereas Aβ1-42 still had no impact on NMDA-induced pressor responses in the presence of ifenprodil, it might rule out the possibility of multiple regulations by Aβ1-42 of NMDA receptor functions in the present study.

Memantine is an uncompetitive antagonist, a potent inhibitor of the NMDA receptors channel, which regulates the cardiovascular system [46,47]. Memantine selectively blocks the excitotoxic effects of glutamate in extrasynaptic neurons, including the Ca^2+^ channel and oxidative stress, while preserving physiological transmission for normal cellular functions [41,48]. Several studies demonstrated that memantine decreased the levels of Aβ in the brain and age-dependent accumulation of Aβ in transgenic animal models of AD [49,50] and the neuroprotective effects of memantine on Aβ induced neurotoxicity [51,52]. The direct effects of memantine may be involved in the mechanisms of the inhibition of Aβ aggregation and induction of Aβ disaggregation by which memantine reduces Aβ deposition in the brain [53]. Similar to the co-administration of ifenprodil, memantine, an NMDA receptor channel inhibitor, blocked the inhibitory effects of Aβ1-40 on NMDA-induced pressor responses when co-administration of Aβ1-40 with memantine. The above results suggested that the inhibitory effects of Aβ1-40 might involve the ion channel binding site on the NMDA receptor. Like ifenprodil, memantine did not change Aβ1-42 effects on NMDA-induced pressor responses in the RVLM.

NMDA receptors are an ionotropic glutamate receptor family consisting of a tetrameric complex made up of GluN1 and GluN3 subunits that bind glycine (or D-serine), as well as GluN2 subunits that bind glutamate and combine in various ways to form the NMDA receptor assembly [54]. The phosphorylation of its subunits can increase the activity of the NMDA receptor, and dephosphorylation can reduce NMDA receptor activity in many neuronal preparations [31]. NMDA receptor functions are regulated by kinases and phosphatases [55,56]. They also impair NMDA receptor activity by regulating the protein levels and localization of NMDA receptor subunits [13]. In our previous study, we observed that changes in the levels of phosphorylated serine 896 on the GluN1 subunit of NMDA receptors might involve the estrogen regulation of ethanol effects on NMDA receptor function in the RVLM [34]. In our present study, immunohistochemistry analysis showed no significant changes in the immunoreactivity of pGluN1-serine896 and GluN1 in the RVLM region during the ICV administration of Aβ1-40. Our findings suggest that the inhibitory effects of Aβ1-40 may not involve the PKC signaling pathways. CKII plays a central role in regulating the GluN2 subunit composition of synaptic NMDA receptors [57]. CKII phosphorylates GluN2B serine 1480 and disrupts the interaction of GluN2B with PSD-95 and SAP102 [58]. Thus, increases in GluN2B phosphorylated status (pGluN2B-serine1480) decreased the surface expression of NMDA receptors. Interestingly, phosphorylation of GluN2B on serine 1480 is regulated by synaptic activity and CaMKII [59]. The immunofluorescence study showed that the Aβ1-40 treatment induced a significant increase in the level of pGluN2B-serine1480 without significant changes in the level of GluN2B in the RVLM. Our results provide the first in vivo evidence that Aβ1-40 phosphorylation of NMDA receptor GluN2B subunits through CKII signaling pathways may play an important role in inhibitory effects of Aβ1-40 on NMDA receptor function in the RVLM. This result suggests that the inhibitory effects of Aβ1-40 might be related to the changes in pGluN2B-serine1480 in the RVLM. However, changes in many signaling pathways and phosphorylated sites on NMDA receptor subunits might modulate NMDA receptor function. Though our results indicated that pGluN2B-serine1480 is an important mechanism involved in Aβ1-40 inhibition of NMDA-induced pressor responses in the RVLM, we cannot rule out the possibility of the involvement of other phosphorylated sites.

In addition to the effects of phosphorylation, Aβ promotes synaptic depression via NMDA receptors acting in a metabotropic rather than ionotropic fashion in the hippocampus slices study [60]. On the other hand, Aβ can also regulate the intracellular trafficking of NMDA receptor subunits or internalize these subunits to affect the surface expression of NMDA receptors, resulting in reduced NMDA receptor activity [13]. Our current immunohistochemistry results did not find that Aβ changes the number of immunoreactive GluN1 or GluN2B neurons in the RVLM. However, we cannot exclude the possibility that using different experimental methods, we may find that Aβ can regulate the surface expression of NMDA receptors via specific mechanisms. Besides phosphorylating and internalizing NMDA receptor subunits [13,38], Aβ can directly affect NMDA channel activity [17]. Thus, Aβ impact on NMDA receptors might involve direct and indirect mechanisms [6].

We quantified the level of human Aβ in rat CSF following ICV administration. Interestingly, the study revealed a much higher level of human Aβ1-40 compared to human Aβ1-42 in the CSF 10 min after ICV administration of the same dose (2 nmol) of the Aβ using an ELISA method. The difference between the two peptides in the CSF may contribute to their differential effects on NMDA receptor function. Different fragments of amyloid peptides have diverse structures. Thus, the aggregation and degradation behaviors might be different from each other [61,62]. Aβ1-42 is more fibrillogenic than Aβ1-40 [63]. A study showed that Aβ1-42 oligomers might bind more to GM1 ganglioside than Aβ1-40, which causes a much more rapid sequestration of the peptides from brain interstitial fluid than monomers [64]. Thus, the degradation, aggregation, and adhesion to the cells in the CSF might differ between Aβ 1-40 and Aβ 1-42, contributing to the different significant levels of the peptides following ICV same dose administration. Before the ICV injection of Aβ1-40 and Aβ1-42, we detected the concentrations of Aβ1-40 and Aβ1-42 using ELISA and found that they were comparable. The present study applied 2 nmol (0.4 mM, 5 μL) of peptides by ICV injection, and the results from ELISA data showed the concentration of the peptides in the CSF was less than 100 nM 10 min after the application, suggesting that the elimination of both peptides might be ongoing following the administration in the brain. Based on the results, however, Aβ1-42 might be eliminated much faster than Aβ1-40, causing significantly lower levels of Aβ1-42. In addition, the present in vivo study also showed that Aβ1-40 modulated NMDA receptor function at a concentration comparable to or lower than the effective concentration of soluble Aβs reported in several in vitro studies [6,13,38,60]. Different potencies of soluble Aβs may result from diverse experimental conditions, probably resulting in different sensitivity of the receptors. Further studies would be required to clarify the mechanism underlying this difference.

The present findings that Aβ1-40 inhibits NMDA receptor function through changed the phosphorylation status pGluN2B-serine1480 of NMDA receptors, which is uniquely phosphorylated by CKII signaling pathways, in the RVLM, might provide one of the mechanistic explanations for Aβ1-40 decreases in NMDA receptor activity underlying neurotoxicity in other brain areas. In addition, the inhibitory effects of Aβ1-40 on NMDA receptor function in the RVLM neurons play an important role in the central control of sympathetic tone and cardiovascular functions.

## 5. Conclusions

In conclusion, the present results in vivo showed that intracerebroventricular administration of Aβ1-40 inhibited N-methyl-D-aspartate (NMDA) receptor function in the rostral ventrolateral medulla; Aβ1-42 at the same dose as Aβ1-40 did not produce noticeable effects on NMDA receptor function. Treatment with an NMDA channel blocker and an NMDA GluN2B receptor antagonist blocked the inhibitory effects of Aβ1-40. An increase in phosphoserine 1480 of the GluN2B subunit (regulated by casein kinase II) might be related, at least partly, to the inhibitory effects of Aβ1-40. The different elimination of the peptides in the cerebrospinal fluid, causing a significant difference in the level of Aβ1-40 and Aβ1-42, might contribute to the differential effects of both peptides on NMDA receptor function.

## Figures and Tables

**Figure 1 biomolecules-13-01736-f001:**
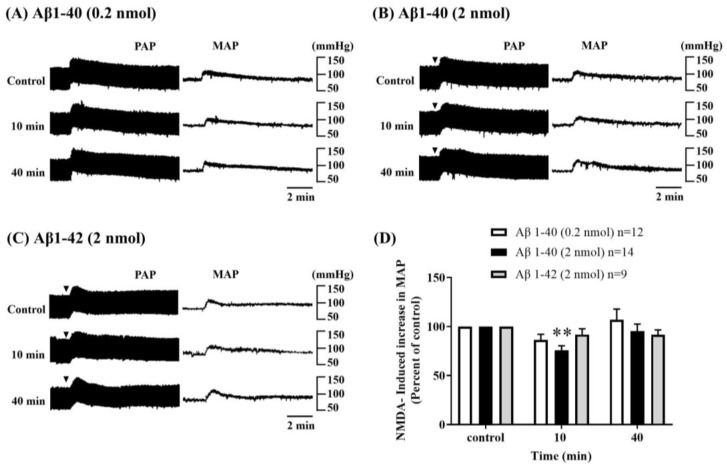
Effect of intracerebroventricular (i.c.v.) injection of amyloid-beta (Aβ) on NMDA receptor function in the RVLM. Representative recordings showed the effects of i.c.v. administration Aβ1-40 (0.2 nmol) (**A**), Aβ1-40 (2 nmol) (**B**), and Aβ1-42 (2 nmol) (**C**) on pulsatile arterial pressure (PAP) and mean arterial pressure (MAP) induced by microinjection of NMDA (0.14 nmol) into the RVLM every 30 min. (**D**) The bar graph shows the percentage changes in NMDA-induced increases in MAP 20 min before (control, i.e., 100%) and 10, 40 min after i.c.v. injection of Aβ1-40 (0.2 nmol, *n* = 12), (2 nmol, *n* = 14), and Aβ1-42 (2 nmol, *n* = 9). Values are mean + SEM. ** *p* ≤ 0.01 compared to the corresponding control analyzed using one-way ANOVA repeated measures followed by Bonferroni’s multiple comparison post-test.

**Figure 2 biomolecules-13-01736-f002:**
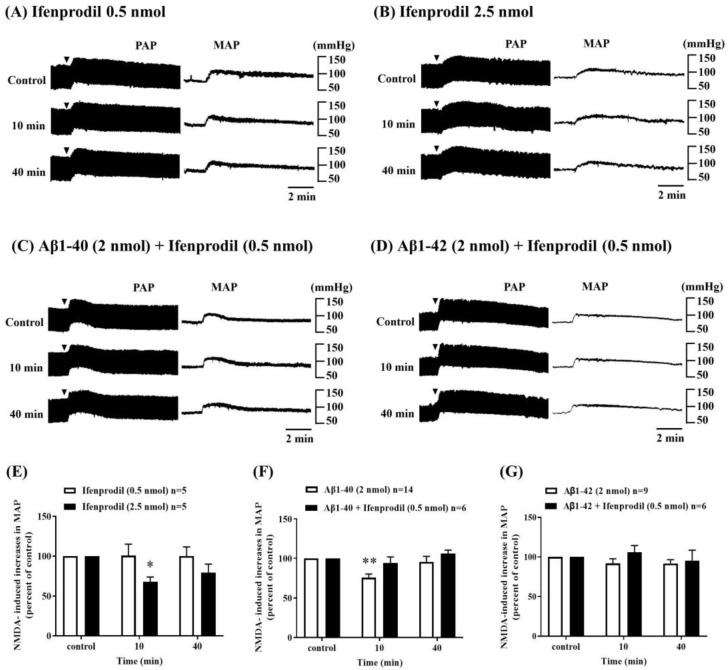
Effect of intracerebroventricular (i.c.v.) co-injection of amyloid-beta (Aβ) with ifenprodil, a GluN2B receptor antagonist, on NMDA receptor function in the RVLM. Representative recordings showed the effects of i.c.v. administration of ifenprodil alone at doses of 0.5 nmol (**A**) and 2.5 nmol (**B**), co-administration of ifenprodil with Aβ1-40 (**C**) and co-administration of ifenprodil with Aβ1-42 (**D**) on increases in pulsatile arterial pressure (PAP) and mean arterial pressure (MAP) induced by microinjection of NMDA (0.14 nmol) into the RVLM. NMDA was applied every 30 min. (**E**) The bar graph shows the percentage changes in NMDA-induced increases in MAP by i.c.v. injection of ifenprodil (0.5 nmol, *n* = 5 and 2.5 nmol, *n* = 5), (**F**) co-administration of Aβ1-40 (2 nmol) with ifenprodil (0.5 nmol), *n* = 6, and (**G**) co-administration of Aβ1-42 (2 nmol) with ifenprodil (0.5 nmol), *n* = 6, 20 min before (control, i.e., 100%) and 10, 40 min after application of NMDA. Values are mean + SEM. * *p* ≤ 0.05, and ** *p* ≤ 0.01 compared to the corresponding control analyzed using one-way ANOVA repeated measures followed by Bonferroni’s multiple comparison post-test.

**Figure 3 biomolecules-13-01736-f003:**
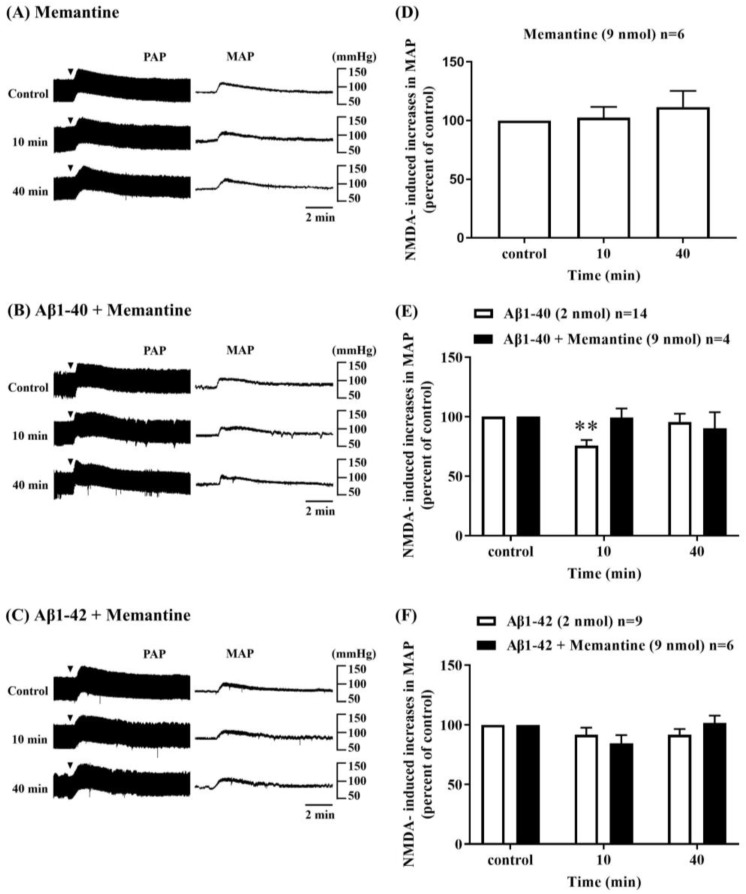
Effect of intracerebroventricular (i.c.v.) co-injection of amyloid-beta (Aβ) with memantine, an NMDA channel blocker, on NMDA receptor function in the RVLM. Representative recordings showed the effects of i.c.v. administration of memantine (9 nmol, *n* = 6) alone (**A**), co-administration of memantine with Aβ1-40 (**B**), and co-administration of memantine with Aβ1-42 (**C**) on increases in pulsatile arterial pressure (PAP) and mean arterial pressure (MAP) induced by microinjection of NMDA (0.14 nmol) into the RVLM. NMDA was applied every 30 min. (**D**) The bar graph shows the percentage changes in NMDA-induced increases in MAP by i.c.v. injection of memantine (9 nmol, *n* = 6), (**E**) co-administration of Aβ1-40 (2 nmol) with memantine (9 nmol), *n* = 4 and (**F**) co-administration of Aβ1-42 (2 nmol) with memantine (9 nmol) *n* = 6, 20 min before (control, i.e., 100%) and 10, 40 min after application of NMDA. Values are mean + SEM. ** *p* ≤ 0.01 compared to the corresponding control analyzed using one-way ANOVA repeated measures followed by Bonferroni’s multiple comparison post-test.

**Figure 4 biomolecules-13-01736-f004:**
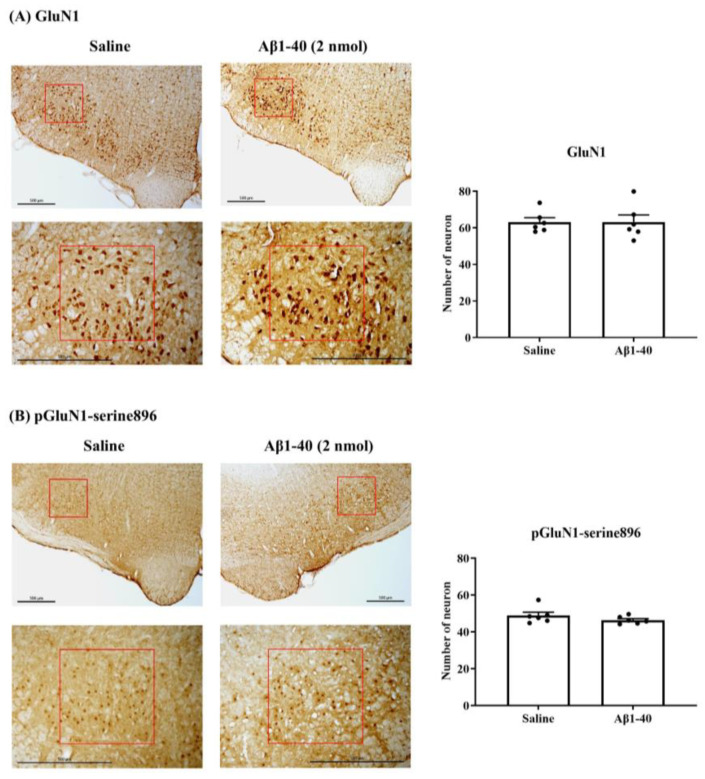
Representative immunohistochemical images show immunoreactive GluN1 (**A**, **left**) or pGluN1-serine 896 (**B**, **left**) neurons in the ventral brainstem at low or high magnification 10 min after saline (5 μL) or Aβ1-40 (2 nmol/5 μL) ICV treatment. The red box refers to the RVLM. The scale bar is 500 μm. The bar graph showed the number of immunoreactive GluN1 (**A**, **right**) or pGluN1-serine 896 (**B**, **right**) neurons in the RVLM after saline or Aβ1-40 treatment. Values are mean + SEM.

**Figure 5 biomolecules-13-01736-f005:**
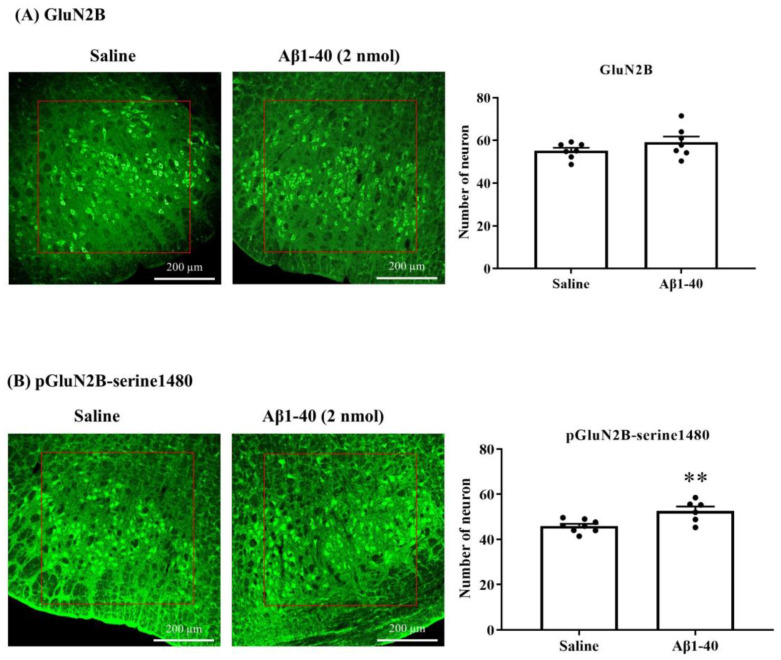
Representative immunofluorescence (IF) images show immunoreactive GluN2B (**A**, **left**) or pGluN2B-serine1480 (**B**, **left**) neurons in the ventral brainstem 10 min after saline (5 μL) or Aβ1-40 (2 nmol/5 μL) ICV treatment. The red box refers to the RVLM. The scale bar is 200 μm. The bar graphs showed the number of GluN2B-IF (**A**, **right**) or pGluN2B-serine1480-IF (**B**, **right**) neurons in the RVLM after saline or Aβ1-40 treatment. Values are mean + SEM. ** *p* ≤ 0.01 compared to the saline group analyzed using unpaired t-test.

**Figure 6 biomolecules-13-01736-f006:**
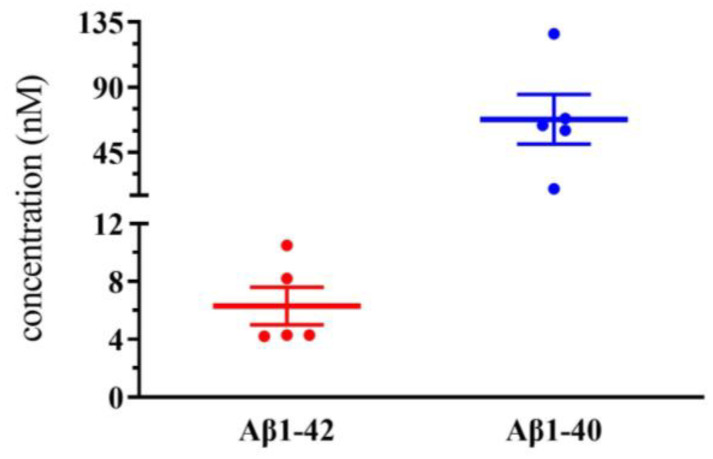
The graph shows the concentration of Aβ1-42 and Aβ1-40 in cerebrospinal fluid 10 min after intracerebroventricular injection of both peptides. Values are mean ± SEM. *n* = 5 for each group. There is a significant difference between the two groups analyzed using unpaired *t*-test.

## Data Availability

Data available upon reasonable request.

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
