# Peer review of "Inhibition of NMDA Receptor Activation in the Rostral Ventrolateral Medulla by Amyloid-β Peptide in Rats"

_biomolecules, 2023, doi:10.3390/biom13121736_

Round 1

Reviewer 1 Report

Comments and Suggestions for Authors

The study provides interesting results showing that the amyloid beta peptide Aβ1-40 produces an inhibitory effect on NMDA receptor function in the rostral ventrolateral medulla, which is responsible for blood pressure control. The Aβ1-40-40 effect was accompanied by serine1480 phosphorylation, which can be a part of the Aβ1-40 inhibition of NMDA receptor function.

The authors offer correct experimental design and data analysis.

However, in discussion, some publications on the topic may be mentioned:

In the study by Tamburri et al., 2013 (10.1371/journal.pone.0065350), the Aβ in the hippocampus induced LTD-related metabotropic signalling, leading to the chronic induction of synaptic depression due to rapid GluN1 internalization. In the current study on RVLM neurons, no effects on GluN1 expression were found. What can be the source of this difference?

Why Aβ1-40 effect lasts only 10 min and vanishes at 40 min time point?

Minor comments:

Line 75: better use GluN2A instead of NR2A for uniform receptor nomenclature

Line 354: “treatment of Aβ1-40 with ifenprodil” – co-application of Aβ1-40 with ifenprodil ?

Line 393: levels of phosphorylated serine 896

Reviewer 2 Report

Comments and Suggestions for Authors

The authors studied the effects of amyloid-beta peptides that are hallmarks of Alzheimer's disease (Abeta(0-40) and Abeta(1-42)) on the N-methyl-D-aspartate (NMDA) receptor inactivation.
They directly measure the alteration of pressure in the hub of blood circulation of the central nervous system (CNS), upon different conditions (sections 3.1-3.3).
Then, they study the effect of Abeta(1-40) on phosphorylation pathways that acts on NMDA function at the level of its subunits (sections 3.4-5).
Finally, they attribute the larger in vivo effects of Abeta(1-40) compared to Abeta(1-42) to the faster degradation of the latter (section 3.6).

This is undoubtely an original set of experiments, that tackles the very interesting relationship between the break of ordinary keeping of SNC and neurodegeneration.
Though the study of interactions of Abeta peptides with NMDA receptor is known, the direct measurement of pressure in a specific critical compartment is new.

Major issues are below.
1) the way the MAP data are elaborated;
2) the validitiy of ELISA assay on Abeta peptides under aggregation in CSF.

1) In Fig.1A we see one single experiment of MAP in case of Abeta(1-40) 0.2 nmol injection.
In Fig.1D, white bars, the corresponding 40 minutes observation is reported as an average of 12 replicates.
It looks that many experiments are required, but there is no SEM in control.
Also, there is strong dependence from the time sampled.
The analysis deserves more comments on robustness, because the effect is small and error is large.

2) There is debate in the literature about ELISA assay for Abeta(1-42), that undergoes fast oligomerization and aggregation (order of 1-2 hours, depending on concentration and on many parameters).
Also, there may be effects of used doses.
In many in vitro experiments mimicking in vivo conditions, the concentration is assumed in the micro M range.
Other microinjection experiments reported in the literature use different doses.
In Fig.6 we see nM observed concentration 10 minutes after injection.
It looks like the injection is at high doses, compared to other experiments.
Comments are required about the microinjection conditions and, when possible, different doses should be compared with care.

Minor points are below.

The abstract is too rich in details.
Conversely, Conclusion section is very coincise, so that it resembles the abstract..
Acronyms should not be used in the summary.

Introduction -
Introduce NMDA.
Introduce a list of all used (many) acronyms.

Section 3.6 -
ELISA assay is really sensitive to any form of Abeta, including oligomers and protofibrils?
This issue should be discussed in more details.

Abeta(1-40) seems to interact more efficiently with NMDA receptor than Abeta(1-42).
There are many hypotheses in the literature about the mechanism (direct vs indirect) of interactions.
The Discussion section should include those.

Comments on the Quality of English Language

Only a few sentences are not clear.

Round 2

Reviewer 2 Report

Comments and Suggestions for Authors

The authors inserted into the manuscript all of the required explanations.

The manuscript is suitable to publication in Biomolecules.